

# CVG-Net: novel transfer learning based deep features for diagnosis of brain tumors using MRI scans

Shaha Al-Otaibi[1], Amjad Rehman[2], Ali Raza[3], Jaber Alyami[4] and Tanzila Saba[2]

[1] Department of Information Systems, Princess Nourah bint Abdulrahman University, Riyadh, Saudi Arabia
[2] Artificial Intelligence & Data Analytics Lab CCIS, Prince Sultan University, Riyadh, Saudi Arabia
[3] Institute of Computer Science, Khwaja Fareed University of Engineering and Information Technology, Rahim Yar Khan, Pakistan
[4] Department of Diagnostic Radiology, Faculty of Applied Medical Sciences, King Abdulaziz University, Jeddah, Saudi Arabia

Corresponding author
Amjad Rehman, arkhan@psu.edu.sa

## ABSTRACT

Brain tumors present a significant medical challenge, demanding accurate and timely diagnosis for effective treatment planning. These tumors disrupt normal brain functions in various ways, giving rise to a broad spectrum of physical, cognitive, and emotional challenges. The daily increase in mortality rates attributed to brain tumors underscores the urgency of this issue. In recent years, advanced medical imaging techniques, particularly magnetic resonance imaging (MRI), have emerged as indispensable tools for diagnosing brain tumors. Brain MRI scans provide high-resolution, non-invasive visualization of brain structures, facilitating the precise detection of abnormalities such as tumors. This study aims to propose an effective neural network approach for the timely diagnosis of brain tumors. Our experiments utilized a multi-class MRI image dataset comprising 21,672 images related to glioma tumors, meningioma tumors, and pituitary tumors. We introduced a novel neural network-based feature engineering approach, combining 2D convolutional neural network (2DCNN) and VGG16. The resulting 2DCNN-VGG16 network (CVG-Net) extracted spatial features from MRI images using 2DCNN and VGG16 without human intervention. The newly created hybrid feature set is then input into machine learning models to diagnose brain tumors. We have balanced the multi-class MRI image features data using the Synthetic Minority Over-sampling Technique (SMOTE) approach. Extensive research experiments demonstrate that utilizing the proposed CVG-Net, the k-neighbors classifier outperformed state-of-the-art studies with a k-fold accuracy performance score of 0.96. We also applied hyperparameter tuning to enhance performance for multi-class brain tumor diagnosis. Our novel proposed approach has the potential to revolutionize early brain tumor diagnosis, providing medical professionals with a cost-effective and timely diagnostic mechanism.

# INTRODUCTION

Brain tumors are abnormal growths of cells within the brain that can disrupt normal brain function and pose serious health risks (*Miao et al., 2023*). Detecting and diagnosing brain tumors is paramount for timely intervention and treatment. MRI brain scans have become a crucial tool (*Zhu et al., 2023*). MRI scans utilize powerful magnets and radio waves to generate detailed images of the brain structure, allowing healthcare professionals to visualize abnormalities (*Liu et al., 2023b*). The MRI scans are especially effective in identifying brain tumors because they provide high-resolution images that can differentiate between healthy brain tissue and tumor masses.

Brain tumors can be classified into various types, each with its distinct characteristics and potential challenges in diagnosis and treatment (*Mehnatkesh et al., 2023*; *Liu et al., 2023a*). Glioma tumors originate in the glial cells, which provide support and protection to neurons (*Oztek et al., 2023*). They are often classified according to their aggressiveness, with high-grade gliomas being particularly concerning due to their rapid growth and infiltrative nature. On the other hand, meningioma tumors develop from the meninges, the protective membranes surrounding the brain and spinal cord. While often benign, they can cause symptoms by compressing nearby brain structures (*Bhatele & Bhadauria, 2023*). Pituitary tumors, which arise in the pituitary gland located at the base of the brain, can affect hormone production and regulation, leading to a wide range of health issues.

The mortality rates caused by brain tumors are increasing daily. Between 2014 and 2018, the average annual age-adjusted incidence rate for primary malignant brain tumors was 7.06 per 100,000 individuals. Projections indicated that in 2022, an estimated 25,050 new diagnoses and 18,280 fatalities would be attributed to primary malignant brain tumors (*Thierheimer et al., 2023*). Brain tumors were 24.25 per 100,000 individuals, comprising 7.06 per 100,000 for malignant cases and 17.18 per 100,000 for non-malignant cases (*Li et al., 2023a*). Similarly, based on the 2018 data from the Global Cancer Registry, a total of 18,078,957 cancer cases were reported across all genders, with 29,681 cases attributed to brain cancer (*Khazaei et al., 2020*).

Traditional diagnostic methods for brain tumors have primarily relied on radiological imaging techniques such as computed tomography (CT) (*Zhang et al., 2023*) and MRI. Classical treatment modalities for brain tumors encompass surgical resection, radiation therapy, and chemotherapy, with the specific approach chosen depending on tumor type, location, and patient characteristics. Deep learning based methods use MRI images analyzed using convolutional neural networks (CNN) for brain tumor recognition (*Rajinikanth, Kadry & Nam, 2021*; *Rajinikanth et al., 2022*; *Maqsood, Damasevicius & Shah, 2021*; *Maqsood, Damaševičius & Maskeliūnas, 2022*; *Badjie & Deniz Ülker, 2022*; *Kurdi et al., 2023*; *Khan et al., 2023*).

In this study, we propose an advanced image-processing mechanism that effectively diagnoses brain tumors without human intervention. We introduce a novel neural network, CVG-Net, for transfer learning-based spatial feature extraction from MRI images. These newly created transfer features are then employed to diagnose brain tumors using machine learning models.

Our contributions regarding brain tumor diagnosis are as follows:

- We have introduced a novel neural network method, CVG-Net, for feature engineering. This method combines 2DCNN and VGG16 to extract spatial features from MRI images. The resulting hybrid spatial feature set is input into machine learning models to diagnose brain tumors.
- We employed the SMOTE technique to balance the multi-class MRI image feature data, resulting in more accurate and reliable diagnoses of brain tumors. This approach addresses class imbalance issues within medical image analysis.
- We utilized four advanced machine learning models and two deep learning models based on neural networks for comparative performance evaluations. Each method's performance is optimized through hyperparameter tuning and validated using k-fold cross-validation. Additionally, we calculated the computational complexity of each method for diagnosing brain tumors.

The rest of the study is organized as follows: 'Literature analysis' analyzes the limitations in previous literature. 'Proposed Methodology' demonstrates our proposed research methodology for diagnosing brain tumors. 'Results and discussions' comparatively evaluates the results of the applied neural network approaches. Our findings are concluded in 'Conclusions and future work'.

## LITERATURE ANALYSIS

This literature analysis section comprehensively examines existing studies and methodologies related to both neural network applications in medical imaging and transfer learning techniques. It delves into the foundational work in medical image analysis, emphasizing the pivotal role of neural networks in automating the detection and diagnosis of various pathologies, including brain tumors. Additionally, it synthesizes the key findings and methodologies from relevant studies, providing a robust foundation for the subsequent implementation of transfer learning techniques in the proposed diagnosis framework for brain tumors using MRI scans

### Machine learning models analysis

The study *Stadlbauer et al. (2022)* investigates the potential of employing multiclass machine learning (ML) algorithms on an extensive array of radiomic features derived from advanced MRI (advMRI) and physiological MRI (phyMRI) data. This innovative methodology, referred to as radiophysiomics, aims to achieve precise classification of contrast-enhancing brain tumors. The study leveraged a substantial MRI database, encompassing more than 1700 MR examinations of brain tumor patients, for the purpose of model development. The findings revealed that the application of adaptive boosting and random forest algorithms in conjunction with both advMRI and phyMRI data surpassed human reading in various performance metrics, including accuracy (0.875 *vs.* 0.850), precision (0.862 *vs.* 0.798), and F-score (0.774 *vs.* 0.740). Nevertheless, it was observed that radiologists demonstrated higher sensitivity (0.767 *vs.* 0.750) and specificity (0.925 *vs.* 0.902) compared to the machine learning models.

The study *Latif et al. (2022)* introduces an innovative method for categorizing glioma tumors, utilizing features based on deep learning in conjunction with a support vector machine (SVM) classifier. Through the application of a deep convolutional neural network for feature extraction from MR images sourced from the BraTS dataset, the study achieved noteworthy levels of accuracy in classification. Specifically, an accuracy of 96.19% was observed for the HGG Glioma type using the FLAIR modality, and a commendable accuracy of 95.46% was obtained for the LGG glioma tumor type when employing the T2 modality. Notably, this method encompassed the classification of four distinct glioma classes, including edema, necrosis, enhancing, and non-enhancing. This pioneering approach shows promising potential in the domain of glioma tumor classification.

## Deep neural networks models analysis

This study *Archana, Karthigha & Suresh Lavanya (2023)* introduces a comparative analysis of several optimizers employed in CNNs for the purpose of brain tumor detection. Within deep learning (DL), artificial neural networks (ANNs) play a crucial role in discerning brain tumors. The medical image dataset is employed for applied methods training and evaluation. Among the various approaches investigated, the application of the AlexNet architecture in conjunction with the stochastic gradient descent (SGD) optimizer demonstrated superior performance. This proposed approach achieved a diagnostic accuracy of 0.80.

This article *Sharif et al. (2020)* introduces a novel approach to actively employ deep learning for the segmentation and recognition of brain tumors in MRI images. The authors utilize the pre-trained Inception V3 CNN model for robust deep feature extraction during the classification phase. These features are integrated with the dominant rotated local binary pattern (DRLBP) to enhance texture analysis. Subsequently, a particle swarm optimization (PSO) technique is employed to fine-tune the concatenated feature vector for classification using a softmax classifier. The study is conducted in two phases. In the initial phase, the segmentation approach (SbDL) is validated on BRATS2017 and BRATS2018 datasets, demonstrating promising Dice scores. For instance, the BRATS2017 dataset achieves Dice scores of 83.73% for the core tumor, 93.7% for the whole tumor, and 79.94% for the enhanced tumor. Moreover, the BRATS2018 dataset attains a Dice score of 88.34%. In the subsequent phase, the classification strategy is applied to BRATS2013, 2014, 2017, and 2018 datasets, yielding an average accuracy of over 92%.

The study *Kibriya et al. (2022)* is focused on developing an automated method for precise brain cancer identification through tumor diagnosis. Previous systems have struggled with issues related to subpar accuracy and high false-positive rates. To address this, the study introduces a pioneering 13-layer CNN architecture specifically designed for classifying brain tumors from MRI scans. The effectiveness of the proposed model was assessed using a standardized dataset comprising 3064 MRI images encompassing three distinct categories of brain cancer: glioma, pituitary, and meningioma. The results demonstrated an average accuracy of 97%.

The study *Fabelo et al. (2019)* introduces a novel framework centered on deep learning techniques, tailored for the analysis of hyperspectral images depicting live human brain

tissue. Through rigorous evaluation using a diverse database of 26 *in vivo* hyperspectral cubes from 16 individual patients, encompassing a total of 258,810 annotated pixels, the framework proves adept at generating a thematic representation that delineates the brain's parenchymal area and accurately identifies the tumor's position. This information serves as a crucial aid to surgeons, significantly enhancing the precision of tumor resection procedures. Notably, the deep learning pipeline 2D convolutional neural network exhibits an impressive overall accuracy of 80% for multiclass classification, which is low compared to baseline.

## Hybrid models analysis

The author of this *Sudharson et al. (2022)* study proposed a hybrid deep learning-based neural system for the detection of brain tumors. The study introduced fundamental concepts of image separation to address the challenges associated with partitioning brain MRI images. Various MRI pre-processing techniques were then elaborated, encompassing tasks such as image registration, bias field correction, and non-brain tissue removal. Additionally, the author presented a hybrid CNN-based partitioning approach, which proved advantageous for delineating brainstem tumors in MRI scans when augmented with prior knowledge. The proposed approach demonstrated an accuracy score of 0.93.

The study *Yazdan et al. (2022)* introduces an innovative approach to diagnosing brain tumors. It employs a dual-pronged strategy, utilizing a multi-scale CNN (MSCNN) architecture to develop a robust classification model, while also addressing the impact of Rician noise on MSCNN performance. This model enables the multi-classification of MRIs, distinguishing between glioma, meningioma, pituitary, and non-tumor cases. Additionally, a denoising process is implemented using a fuzzy similarity-based non-local means (FSNLM) filter to enhance classification results. The study utilizes an openly accessible MRI dataset comprising 3,264 MRIs. Experimental results demonstrate that the proposed MSCNN model outperforms both AlexNet and ResNet, exhibiting superior accuracy and efficiency while incurring lower computational expenses. Specifically, the MCNN2 iteration achieves an impressive accuracy and F1-score of 91.2% and 91%, respectively.

The analyzed works with their research limitations are summarized in Table 1.

## PROPOSED METHODOLOGY

This section comprehensively describes the materials and methods used to evaluate our research experiments. We examine the MRI scans dataset and the advanced neural network approaches applied. Figure 1 illustrates our novel proposed research methodology for diagnosing brain tumors. The following steps outline the process carried out in our proposed methodology for conducting experiments.

- **Step 1:** At the outset of our research experiments, we acquired a brain tumor-related dataset of MRI scan images for evaluating results. This dataset encompasses multiple target classes, allowing for a comprehensive assessment of our findings.

**Table 1** The literature summary and limitations analysis.

| Ref | Dataset | Proposed technique | Performance accuracy | Research limitations |
|---|---|---|---|---|
| *Archana, Karthigha & Suresh Lavanya (2023)* | Brain tumours medical images | AlexNet with SGD | 0.80 | Low diagnose performance scores |
| *Sudharson et al. (2022)* | Brain MRI image scans | Hybrid CNN | 0.93 | Low performance with a state of art approach |
| *Sharif et al. (2020)* | BRATS2017 and BRATS2018 datasets | Inception V3 pre-trained CNN model | 0.92 | Old datasets with low performance accuracy for diagnosis |
| *Kibriya et al. (2022)* | 3,064 MRI images | 13-layer CNN architecture | 0.97 | Classical neural networks were used for classification with low images data record |
| *Yazdan et al. (2022)* | 3,264 MRIs Scans | Multi-Scale CNN (MSCNN) architecture | 0.91 | Poor accuracy performance for multi class data with low images data record |
| *Stadlbauer et al. (2022)* | 1,700 MRI examinations from advMRI and phyMRI data | Adaptive Boosting and Random Forest | 0.87 | Low diagnose performance scores with low images data record |
| *Fabelo et al. (2019)* | 26 *in vivo* hyperspectral cubes from 16 different patients, among which 258,810 pixels | 2D convolutional neural network | 0.80 | Poor accuracy performance for multiclass classification |
| *Latif et al. (2022)* | BraTS dataset | Support vector machine (SVM) | 0.96 | Classical machine learning approaches were used with outdated segmented images data record |

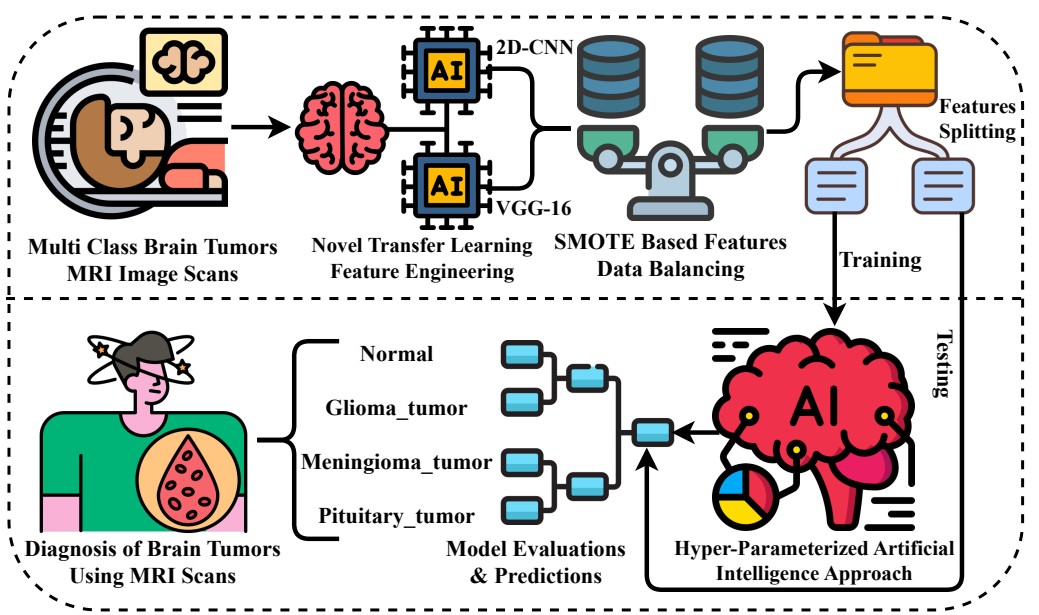

**Figure 1** Workflow diagram of our proposed methodology for brain tumor diagnosis.

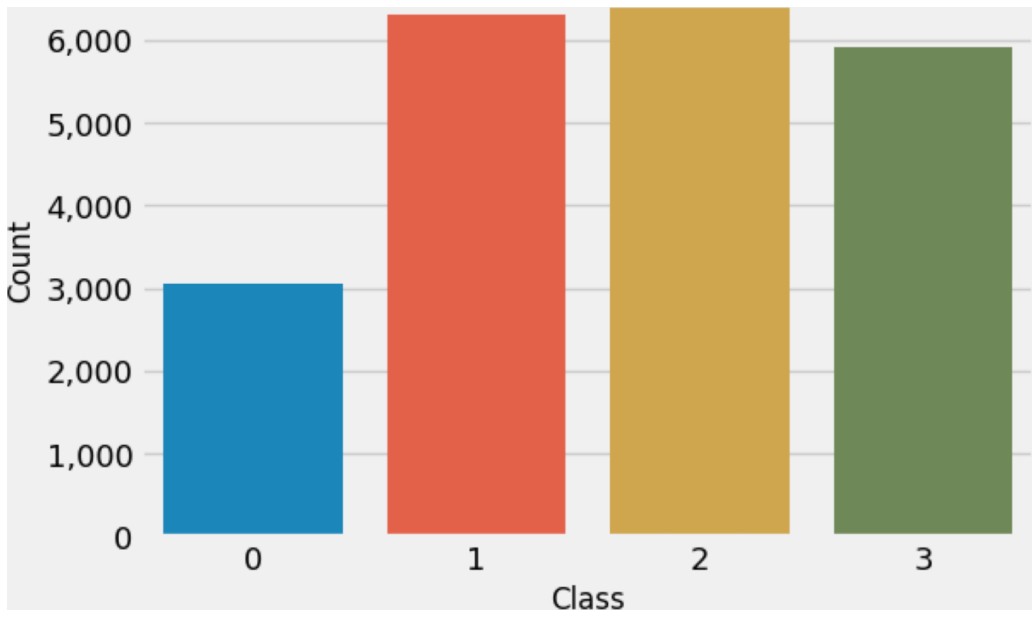

**Figure 2** **The target class labels distributions analysis.**

- **Step 2:** Next, for the extraction of rich-level features from MRI images, we have proposed a novel neural network approach that leverages transfer learning for spatial feature extraction, ultimately creating a new feature set.
- **Step 3:** We then analyzed the dataset and identified an imbalance. To address this issue, we applied the SMOTE approach to balance the features of the dataset for multi-class classification.
- **Step 4:** The image feature data is then divided into two portions: training and testing. The split ratio employed is 80% for training and 20% for testing.
- **Step 5:** We have employed advanced machine learning and deep learning approaches in this step. Using 80% of the training data, we have implemented the applied methods.
- **Step 6:** Then, the 20% portion of the data is utilized for the evaluations and predictions of applied methods through various performance metrics in comparison.
- **Step 7:** Ultimately, the superior neural network method during experiments is employed for brain tumor diagnosis.

## Multi class brain tumors MRI image data

This research used multiclass MRI scan images (*Hashemi, 2023*) related to brain tumors to conduct experiments. The dataset comprises 21,672 MRI scan images. Figure 2 illustrates the distribution of multi-class target labels. This analysis reveals that the 'Normal' target class (0) contains 3,066 samples, the 'Glioma_tumor' target class (1) contains 6,307 samples 'Meningioma_tumor' target class (2) contains 6,391 samples, and the 'Pituitary_tumor'

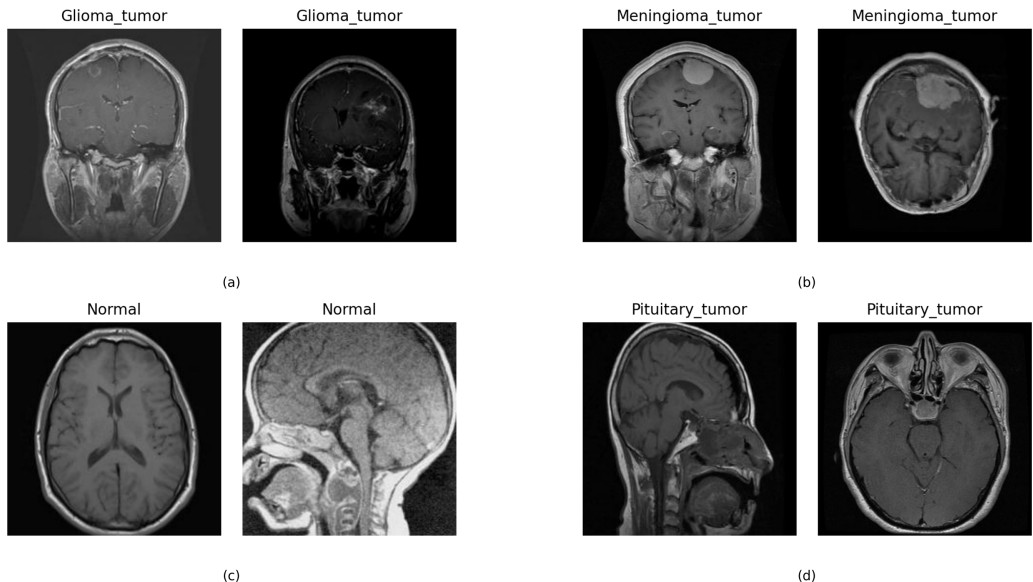

**Figure 3** The sample dataset image analysis with the target class.

target class (3) contains 5,908 samples. Additionally, we conducted a sample image analysis, as shown in Fig. 3.

We utilized the state-of-the-art dataset: Multi-class Brain Tumour MRI Image Data, comprising 21,672 MRI scan images. We chose this dataset because it aligns with the problem addressed in our research study, specifically, brain tumor detection. Moreover, this dataset encompasses a substantial number of images, enhancing the generalizability of our research findings.

## Novel transfer learning approach

This section comprehensively describes our novel proposed CVG-Net approach for transfer feature engineering. The step-by-step architectural workflow of the proposed approach is illustrated in Fig. 4. The brain tumor MRI image data is initially fed into the VGG16 approach, which extracts rich-level spatial features based on transfer learning. Similarly, the brain tumor MRI image data is input into the 2DCNN approach, which also extracts rich-level spatial features using transfer learning. The spatial features obtained from both approaches are then combined to create a new feature set. This newly generated feature set is subsequently employed in building the machine learning models applied in this study and for further results evaluations.

The Algorithm 1 shows the step-by-step workflow of the proposed transfer learning feature engineering.

## SMOTE based data balancing

SMOTE has emerged as a pivotal tool in addressing class imbalance issues within medical image analysis, particularly in diagnosing brain tumors using multi-class MRI image datasets (*Li et al., 2023b*). Class imbalance, where certain classes are underrepresented

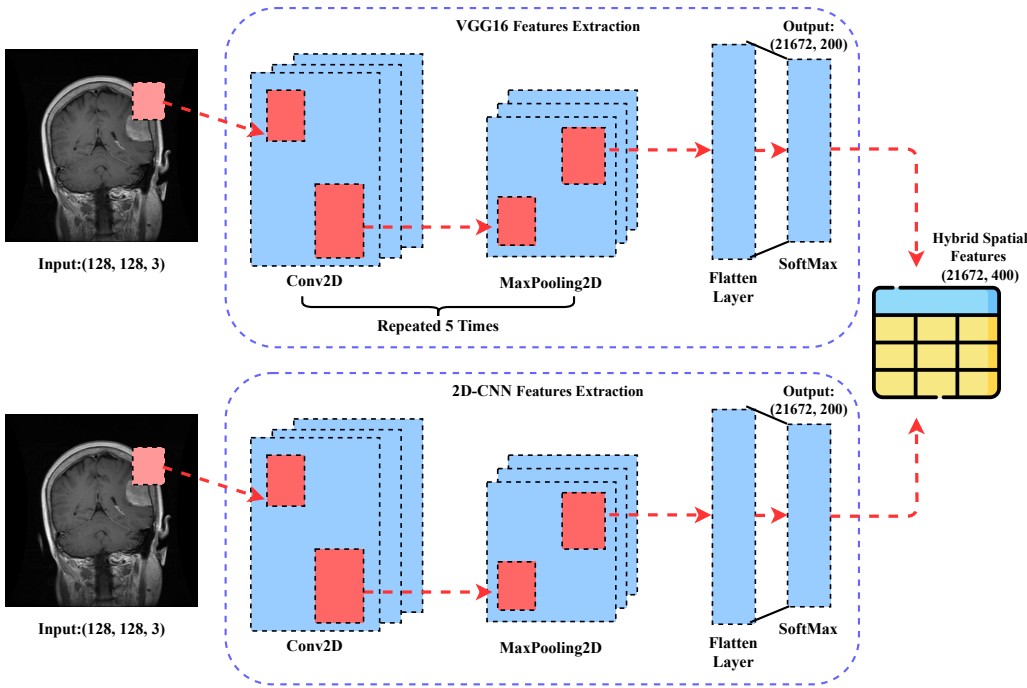

**Figure 4** The architectural workflow analysis of our proposed CVG-Net feature engineering.

---

**Algorithm 1** CVG-Net Algorithm

---

**Input:** Multi-class MRI scans related to brain tumors.

**Output:** New Transfer learning-based deep features.

initiate;

1- $F_{cnn} \longleftarrow CNN_{prediction}(MiS)$  // $MiS \in MRI\ images\ set$,    here $MiS$ is original image data and $F_{cnn}$ is the deep spatial feature set extracted.

1- $F_{vgg16} \longleftarrow VGG16_{prediction}(MiS)$  // $MiS \in MRI\ images\ set$,    here $MiS$ is original image data and $F_{vgg16}$ is the deep spatial feature set extracted.

3- $F_{Spat} \longleftarrow F_{cnn} + F_{vgg16}$  // here $F_{Spat}$ is final hybrid deep spatial features set used for osteoarthritis detection.

end;

---

relative to others, can significantly hinder the performance of machine learning models, as they tend to be biased towards the majority class. In medical imaging, this can have critical implications for patient care. SMOTE operates by generating synthetic samples in the feature space of the minority class, thereby alleviating the imbalance. When applied to multi-class MRI datasets for brain tumor diagnosis, SMOTE ensures that the model is exposed to a more representative data distribution, effectively discriminating between various tumor types. This methodology has demonstrated promising results, contributing to more accurate and reliable diagnoses, paramount in clinical decision-making processes.

## Applied neural network methods

The application of artificial intelligence (AI) based machine learning, deep learning, and transfer learning methods has emerged as a transformative approach in medical imaging (*Rehman et al., 2023*; *Qadri et al., 2023*; *Raza et al., 2022b*), particularly in diagnosing brain tumors using brain MRI scans. These advanced techniques have revolutionized the way medical professionals analyze and interpret complex neuroimaging data. Machine learning algorithms can efficiently extract intricate patterns and features from large datasets of MRI scans, enabling them to differentiate between normal brain tissue and pathological regions associated with brain tumors.

Deep learning, in particular, has been instrumental in automating the tumor detection, segmentation, and classification process, as it can automatically learn hierarchical representations from raw MRI data, making it highly effective in identifying subtle abnormalities (*Abunadi & Kumar, 2021*; *Inbarani H & Azar, 2020*; *Azar et al., 2023*). Transfer learning techniques leverage pre-trained models on large datasets and have shown promise in enhancing diagnostic accuracy by transferring knowledge learned from other domains or medical imaging tasks to improve the performance of brain tumor diagnosis. Integrating AI-driven methods in brain tumor diagnosis accelerates the detection process and aids healthcare providers in making more accurate and timely decisions.

In this research, we employed several advanced machine learning and deep learning-based approaches for performance evaluations. We utilized two neural network approaches, 2D-CNN and VGG16, for both classification and transferring feature mechanisms from MRI images. The machine learning models we employed include the k-neighbors classifier, logistic regression, and random forest. We have provided a detailed description of the working mechanisms of each applied neural network in Table 2 and layer architecture in Table 3.

## Hyperparameter setting

Hyperparameter tuning is a crucial aspect of optimizing neural network models in machine learning. These hyperparameters are configurations external to the model and can significantly impact its performance. In our research, we focus on systematically exploring and optimizing these hyperparameters to enhance the overall efficiency and effectiveness of neural networks. We employ a recursive process of training and evaluations to determine the best-fit hyperparameters. The selected hyperparameters for the neural network applied to diagnose brain tumors are presented in Table 4.

## RESULTS AND DISCUSSIONS

The results and discussions of our study methods proposed for the diagnosis of brain tumors using brain MRI scans are analyzed in this section. We have provided a comprehensive analysis of the results obtained through the application of neural network methods. In this section, we dive into the key findings and their implications for the field of medical imaging and brain tumor diagnosis.

**Table 2  The working mechanism description analysis of applied neural network approaches.**

| Method | Working description |
|---|---|
| 2D-CNN | A 2D-CNN (*Archana, Karthigha & Suresh Lavanya, 2023*; *Zhou et al., 2023*) works by systematically scanning and analyzing small regions (kernels) of a 2D input image, such as a Brain MRI scan. These kernels extract features like edges, textures, and patterns through convolution operations. Multiple layers of convolutions and pooling are used to hierarchically learn and combine these features, enabling the CNN to detect complex patterns indicative of brain tumors, making it an effective method for automated diagnosis using MRI scans. |
| VGG-16 | The VGG-16 neural network method (*Rohith et al., 2023*; *Shourie, Anand & Gupta, 2023*) for diagnosing brain tumors using brain MRI scans operates by utilizing a deep CNN architecture. It employs a series of convolutional layers to extract intricate features from the MRI images, gradually learning hierarchical representations of the input data. These features are then passed through fully connected layers for classification, distinguishing between tumor and non-tumor regions based on learned patterns, ultimately aiding in accurate brain tumor diagnosis. |
| KNC | K-neighbors classifier (KNC) method for diagnosing brain tumors using brain MRI scans operates by analyzing the proximity of each MRI scan to its neighboring data points in a high-dimensional feature space. KNC assigns a class label to a given MRI scan based on the majority class among its k nearest neighbors, where k is a user-defined parameter. This proximity-based approach allows KNC to effectively identify brain tumors by leveraging the collective characteristics of nearby MRI scans in the dataset, making it a valuable tool in medical image analysis (*Raza et al., 2023*; *Raza et al., 2022a*). |
| LR | Logistic regression (LR) (*van Ravesteijn et al., 2010*) for diagnosing brain tumors using MRI scans begins by collecting a dataset of brain MRI images along with corresponding binary labels indicating the presence or absence of tumors. The LR model calculates the probability of a brain tumor being present in each MRI scan by applying a logistic function to a linear combination of image features extracted from the scans. It learns the optimal weights for these features during training. Once trained, the LR model can classify new MRI scans as tumor-positive or tumor-negative based on the probability threshold (typically 0.5), providing a valuable tool for non-invasive brain tumor diagnosis. |
| RF | Random forest (RF) (*Dutta et al., 2023*) for diagnosing brain tumors using MRI scans work by first collecting a diverse dataset of brain MRI images with corresponding tumor labels. The RF algorithm then creates multiple decision trees using random subsets of the data and features, allowing each tree to classify tumor presence or absence independently. RF combines the results from these trees through voting or averaging, resulting in a robust and accurate diagnosis of brain tumors from MRI scans. |

## Experimental setup

Our proposed experimental study setup includes the use of the Google Colab platform to evaluate research results. The platform offers GPU backend support, providing 90 GB of disk space and 13 GB of RAM. We have employed the Python 3.0 programming language to build the neural network used in our research. We utilized the Sklearn, Tensorflow, and
**Table 3** The layered architecture analysis of applied neural network approaches.

| Layer (type) | Output Shape | Param # |
| --- | --- | --- |
| **2D-CNN** | | |
| conv2d (Conv2D) | (None, 126, 126, 64) | 1,792 |
| max_pooling2d (MaxPooling2D) | (None, 63, 63, 64) | 0 |
| dropout (Dropout) | (None, 63, 63, 64) | 0 |
| flatten (Flatten) | (None, 254,016) | 0 |
| dense (Dense) | (None, 4) | 1,016,068 |
| **VGG-16** | | |
| vgg16 (Functional) | (None, 4, 4, 512) | 14,714,688 |
| dropout_2 (Dropout) | (None, 4, 4, 512) | 0 |
| flatten_2 (Flatten) | (None, 8,192) | 0 |
| dense_2 (Dense) | (None, 4) | 32,772 |

**Table 4** The hyper-parameters tuning analysis of applied neural network approaches.

| Method | Hyper-parameters description |
| --- | --- |
| 2D-CNN | activation= 'softmax', Dropout=0.02, optimizer='adam', loss='ategorical_crossentropy', metrics=['accuracy','Precision','Recall'] |
| VGG-16 | weights = 'imagenet', include_top = False, activation= 'softmax', Dropout=0.02, optimizer='adam', loss='categorical_crossentropy', metrics=['accuracy','Precision','Recall'] |
| KNC | n_neighbors=3, weights='uniform', leaf_size=30, $p = 2$ |
| LR | penalty='l2', tol=1e−4, $C = 1.0$, solver='lbfgs' |
| RF | max_depth=50, criterion="gini", random_state=0, n_estimators=50, max_features="sqrt" |

Keras modules for evaluating the results. During performance evaluations, true positives (TP), true negatives (TN), false positives (FP), and false negatives (FN) are used as basic parameters. The performance metrics used for performance evaluation are examined as follows:

- **Accuracy:** in diagnosing brain tumors using brain MRI scans refers to measuring how effectively the diagnostic system or method correctly identifies and classifies the presence or absence of brain tumors in the MRI images.

$$Accuracy = \frac{TP + TN}{TP + TN + FP + FN}. \tag{1}$$

- **Precision:** in diagnosing brain tumors using brain MRI scans refers to measuring how accurately the diagnostic test identifies TP cases among all the positive cases it detects.

$$Precision = \frac{TP}{TP + FP}. \tag{2}$$

- **Recall:** in diagnosing brain tumors using brain MRI scans, is a fundamental performance metric that measures the ability of a diagnostic model or process to correctly identify

and classify TP cases of brain tumors among all the actual cases of brain tumors present in the MRI scans.

$$Recall = \frac{TP}{TP + FN}.$$ (3)

- **F1:** in diagnosing brain tumors using brain MRI scans, is a measure that combines precision and recall to assess the accuracy of the classification model.

$$F1\ Score = \frac{2 \cdot Precision \cdot Recall}{Precision + Recall}.$$ (4)

- **Runtime computations:** is a neural network method's brain tumor diagnosis time by inputting MRI images.
- **Standard deviations (SD):** refers to a statistical measure of the variability of the evaluation metrics obtained from multiple folds of the dataset during the cross-validation process.

$$\sigma = \sqrt{\frac{1}{N} \sum_{i=1}^{N} (x_i - \mu)^2}.$$ (5)

## Performance analysis with classical approaches

The performance analysis of the applied classical deep learning-based approaches, 2DCNN and VGG16, is conducted in this section. MRI brain scan images are fed into the 2DCNN and VGG16 methods, and the results are then evaluated.

### Results of CNN model

The performance results of the applied CNN model during training are illustrated in Fig. 5. We conducted a time series analysis over 20 epochs of the CNN model. The evaluation is based on loss scores, accuracy, precision, and recall. The analysis reveals that the training loss is very high in the first two epochs. Subsequently, there is a notable reduction in loss, while the validation loss gradually increases. Similarly, the training accuracy starts off low in the initial two epochs, but then exhibits a steady increase. The validation accuracy scores fluctuate around 0.80. The precision and recall metrics exhibit a similar pattern to the accuracy performance scores throughout training. In conclusion, this analysis indicates that the CNN model achieved acceptable scores for brain tumor diagnosis during training. Nevertheless, there is still a need for performance enhancement.

The performance analysis of the applied CNN model for unseen testing data is described in Table 5. The analysis shows that the CNN model achieved an acceptable score of 0.76 during brain tumor diagnosis. However, the loss scores are high for the testing data. The class-wise performance scores also demonstrate moderate results. This analysis concludes that classical approaches have not proven fruitful in this case; there is a need for advanced mechanisms for performance enhancement.

### Results of VGG16 model

The performance results of the applied pre-trained VGG16 model during training are illustrated in Fig. 6. We conducted a time series analysis over 20 epochs of the VGG16 model. The analysis reveals that the training loss is very high in the first five epochs.

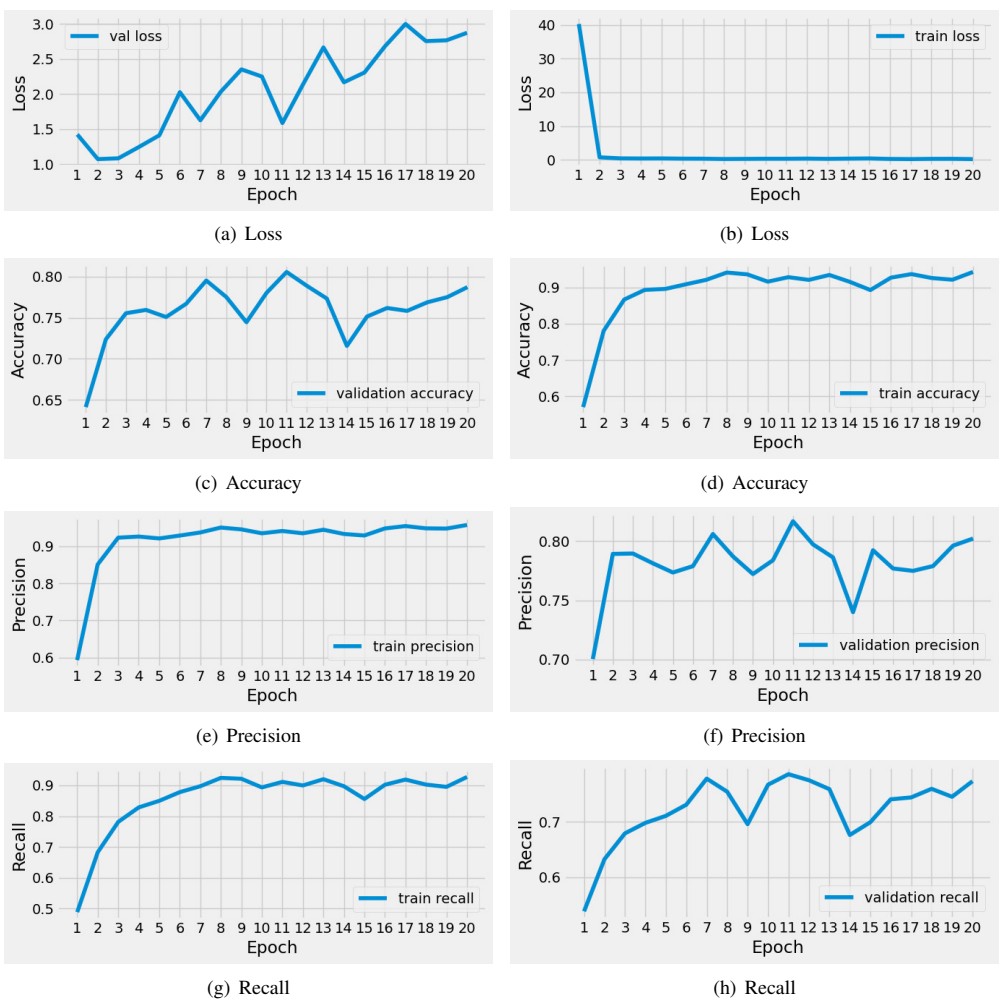

**Figure 5  The time series-based performance results comparison of CNN model during training.** Each performance measure is evaluated during the 20 epochs of training.

**Table 5  The results analysis of applied 2D-CNN neural network.**

| Accuracy | Loss | Target class | Precision | Recall | F1 |
|---|---|---|---|---|---|
| | | Normal | 0.72 | 0.76 | 0.74 |
| | | Glioma_tumor | 0.70 | 0.68 | 0.69 |
| 0.76 | 3.64 | Meningioma_tumor | 0.78 | 0.75 | 0.77 |
| | | Pituitary_tumor | 0.87 | 0.87 | 0.87 |
| | | Average | 0.77 | 0.76 | 0.77 |

Subsequently, there is a notable reduction in loss, while the validation loss gradually increases. Similarly, the training accuracy starts off low in the initial two epochs, but then exhibits a steady increase. The validation accuracy scores fluctuate around 0.92. The precision and recall metrics exhibit a similar pattern to the accuracy performance scores

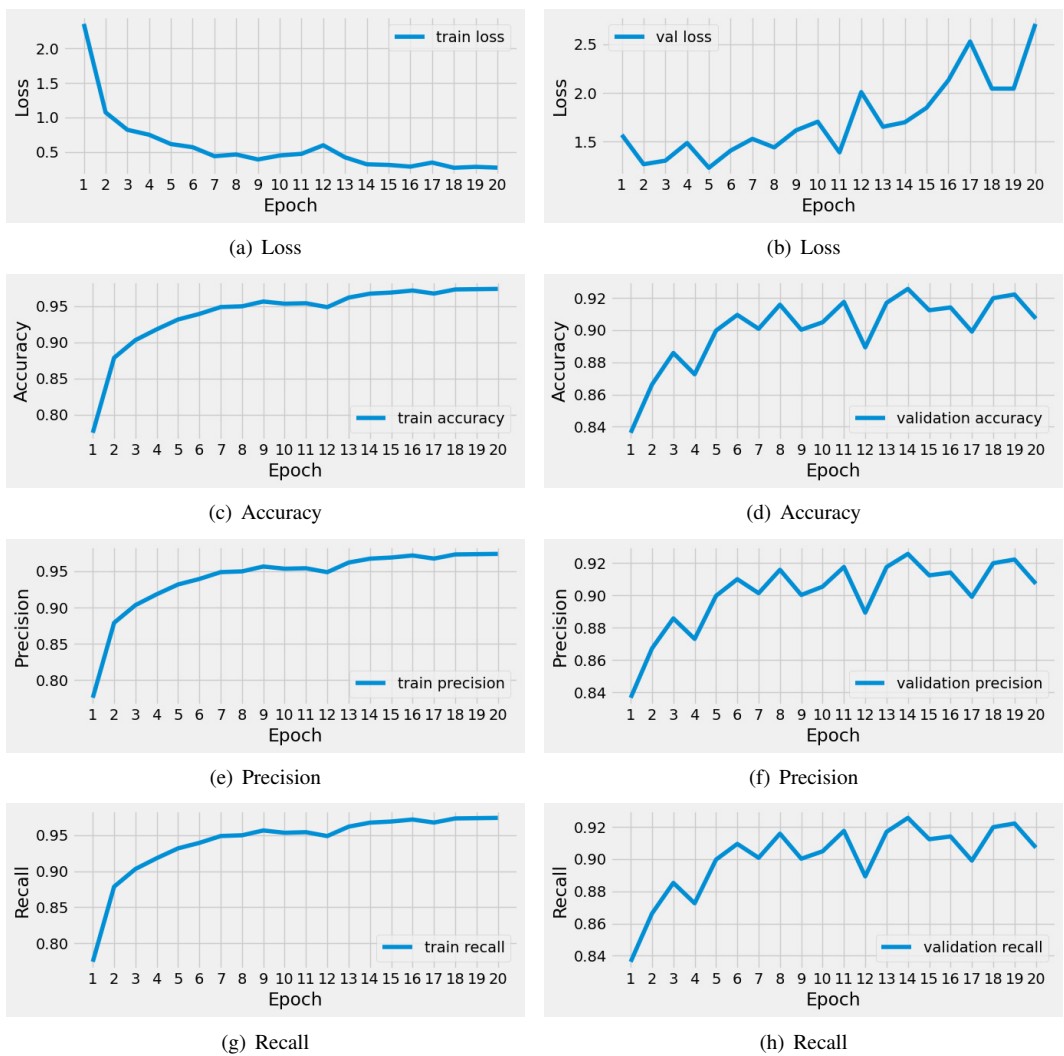

**Figure 6** **The time series-based performance results comparison of VGG16 model during training.**
Each performance measure is evaluated during the 20 epochs of training.

throughout training. In conclusion, this analysis indicates that the VGG16 model achieved good scores for brain tumor diagnosis during training. Nevertheless, there is still a need for performance enhancement.

The performance analysis of the applied VGG-16 model on unseen testing data is outlined in Table 6. The analysis reveals that the VGG-16 model achieved a commendable score of 0.90 in brain tumor diagnosis. Nevertheless, the loss scores were notably high for the testing data. The class-wise performance scores also demonstrate positive results, with 0.91 precision and F1-scores. This analysis leads to the conclusion that classical approaches yielded a moderately satisfactory performance. However, there is a pressing need for advanced mechanisms to enhance performance.

**Table 6  The results analysis of applied VGG-16 neural network.**

| Accuracy | Loss | Target class | Precision | Recall | F1 |
|---|---|---|---|---|---|
| | | Normal | 0.99 | 0.89 | 0.94 |
| | | Glioma_tumor | 0.91 | 0.85 | 0.88 |
| 0.90 | 2.66 | Meningioma_tumor | 0.83 | 0.93 | 0.87 |
| | | Pituitary_tumor | 0.96 | 0.95 | 0.95 |
| | | Average | 0.91 | 0.90 | 0.91 |

**Table 7  The results analysis of applied neural networks with novel unbalanced features.**

| Method | Accuracy | Target class | Precision | Recall | F1 |
|---|---|---|---|---|---|
| | | Normal | 0.87 | 0.81 | 0.84 |
| | | Glioma_tumor | 0.79 | 0.82 | 0.81 |
| LR | 0.81 | Meningioma_tumor | 0.76 | 0.73 | 0.75 |
| | | Pituitary_tumor | 0.87 | 0.91 | 0.89 |
| | | Average | 0.82 | 0.82 | 0.82 |
| | | Normal | 0.95 | 0.85 | 0.90 |
| | | Glioma_tumor | 0.82 | 0.83 | 0.83 |
| RF | 0.85 | Meningioma_tumor | 0.81 | 0.78 | 0.80 |
| | | Pituitary_tumor | 0.87 | 0.95 | 0.91 |
| | | Average | 0.87 | 0.85 | 0.86 |
| | | Normal | 0.94 | 0.93 | 0.94 |
| | | Glioma_tumor | 0.91 | 0.96 | 0.93 |
| KNC | 0.94 | Meningioma_tumor | 0.96 | 0.87 | 0.91 |
| | | Pituitary_tumor | 0.95 | 0.98 | 0.96 |
| | | Average | 0.94 | 0.94 | 0.94 |

## Performance analysis with Novel approach

The performance analysis of the applied advanced machine learning methods with the novel proposed CVG-Net feature engineering is presented in this section. The proposed approach extracts novel transfer features by inputting MRI brain scan images into the 2DCNN and VGG16 methods, creating a hybrid feature set. The newly created features are then input into machine learning models, and the results are subsequently evaluated.

## Results with unbalanced features

In this section, the novel extracted features that are unbalanced are inputted into applied machine learning methods for performance evaluations. The performance results of machine learning methods with unbalanced features are described in Table 7. The analysis demonstrates that applied machine learning models achieved high-performance scores using the novel extracted transfer features. The LR and RF achieved accuracy performance scores of 0.81 and 0.85, respectively. The applied KCN approach achieved a good score of 0.94. This analysis concludes that with unbalanced novel features, good performance is achieved. However, the performance can be further improved by applying advanced data-balancing approaches.

**Table 8** The results analysis of applied neural networks with novel balanced features.

| Method | Accuracy | Target class | Precision | Recall | F1 |
|--------|----------|--------------|-----------|--------|-----|
| LR | 0.84 | Normal | 0.91 | 0.93 | 0.92 |
|  |  | Glioma_tumor | 0.82 | 0.79 | 0.80 |
|  |  | Meningioma_tumor | 0.74 | 0.72 | 0.73 |
|  |  | Pituitary_tumor | 0.87 | 0.90 | 0.89 |
|  |  | Average | 0.84 | 0.84 | 0.84 |
| RF | 0.88 | Normal | 0.96 | 0.97 | 0.96 |
|  |  | Glioma_tumor | 0.83 | 0.83 | 0.83 |
|  |  | Meningioma_tumor | 0.82 | 0.76 | 0.79 |
|  |  | Pituitary_tumor | 0.90 | 0.95 | 0.92 |
|  |  | Average | 0.88 | 0.88 | 0.88 |
| KNC | 0.95 | Normal | 0.96 | 0.99 | 0.97 |
|  |  | Glioma_tumor | 0.93 | 0.95 | 0.94 |
|  |  | Meningioma_tumor | 0.95 | 0.88 | 0.91 |
|  |  | Pituitary_tumor | 0.96 | 0.99 | 0.98 |
|  |  | Average | 0.95 | 0.95 | 0.95 |

## Results with balanced features

After evaluating the performance scores with unbalanced feature data, we applied the famous data balancing Synthetic Minority Oversampling Technique (SMOTE) to the extracted feature data. The evaluation results of the applied machine learning approach with a balanced dataset are demonstrated in Table 8. The analysis reveals that balanced features help improve performance scores for the diagnosis of brain tumors. The proposed KNC method outperformed other applied methods with a high accuracy of 0.95 in comparison. This analysis concludes that SMOTE-based dataset balancing helps us improve performance scores.

The confusion matrix allows visualization of the performance of applied machine learning methods for the diagnosis of brain tumors. The overall performance validation, based on the confusion matrix, is illustrated in Fig. 7. The analysis shows that the applied LR and RF have high wrong predictions for target classes 2 and 3. The minimum error rate, indicating the fewest wrong predictions, is achieved by the proposed K-nearest neighbors classification (KNC) approach. This analysis validates the high-performance scores of the proposed KNC method for the diagnosis of brain tumors using MRI scans.

## Validating performance with k-fold mechanism

We have validated the performance of applied machine-learning approaches in this section using both unbalanced and balanced feature data. The results of k-fold cross-validations are demonstrated in Table 9. To validate accuracy performance results, we used a fold split of 10 and determined the outcomes. The analysis shows that with unbalanced features, LR and RF achieved acceptable validation accuracy scores in the range of 0.81 to 0.84. The proposed approach achieved a commendable score of 0.94. When we evaluated results with the balanced dataset, performance scores increased for each applied approach. The

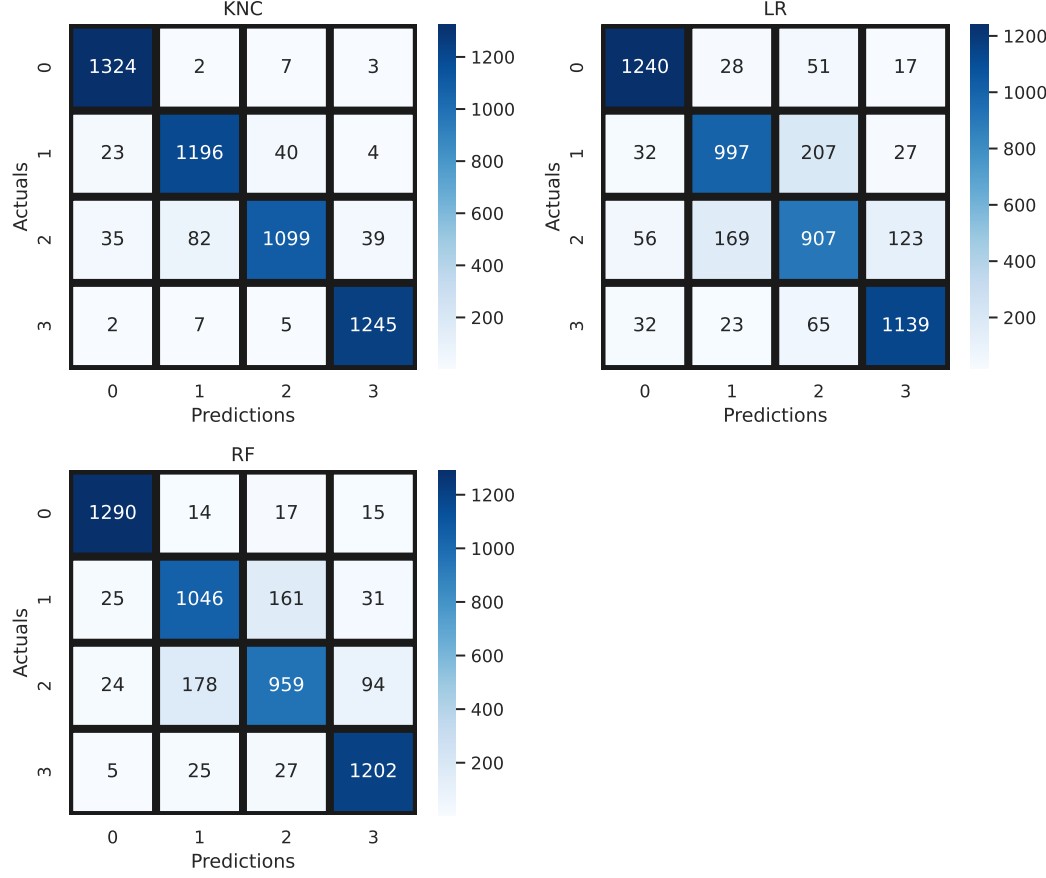

**Figure 7** Confusion matrix analysis of applied neural network approaches.

**Table 9   K-fold cross-validations results analysis of applied neural networks.**

| Method | Unbalanced features | | Balanced features | |
|---|---|---|---|---|
| | 10-fold accuracy | SD (+/-) | 10-fold accuracy | SD (+/-) |
| LR | 0.81 | 0.005 | 0.83 | 0.009 |
| RF | 0.84 | 0.007 | 0.88 | 0.005 |
| KNC | 0.94 | 0.004 | 0.96 | 0.005 |

analysis concludes that the proposed KNC approach achieved the highest k-fold accuracy scores of 0.96 for the diagnosis of brain MRI scans.

## Computational complexity analysis

In this section, we present a computational complexity analysis of applied machine learning methods. The runtime computations, measured in seconds, of these applied machine learning methods, are determined during each model-building process using a newly created features dataset. The computation results are reported in Table 10. Our analysis demonstrates that the proposed KNC method achieved the minimum runtime

**Table 10  Computational complexity analysis of applied neural networks with novel balanced features.**

| Method | Runtime (seconds) |
|---|---|
| LR | 3.7201 |
| RF | 24.215 |
| KNC | 0.0281 |

**Table 11  The performance comparison with state-of-the-art studies for brain tumor diagnosis.**

| Ref. | Year | Proposed technique | Accuracy |
|---|---|---|---|
| *Archana, Karthigha & Suresh Lavanya (2023)* | 2023 | AlexNet with SGD | 0.80 |
| *Sudharson et al. (2022)* | 2022 | Hybrid CNN | 0.93 |
| *Sharif et al. (2020)* | 2020 | Inception V3 with CNN | 0.92 |
| *Yazdan et al. (2022)* | 2022 | Multi-Scale CNN | 0.91 |
| Our | 2024 | CVG-Net-KNC | 0.96 |

computations compared to other applied methods. The proposed KNC method takes only 0.0281 s to diagnose brain tumors.

## Comparison with state-of-the-art studies

For a fair performance comparison, we have included state-of-the-art studies in brain tumor diagnosis in this analysis. The performance comparison of our proposed approach is illustrated in Table 11. The studies considered for performance comparison are from the years 2020, 2022, and 2023. The analysis reveals that our proposed approach outperformed state-of-the-art studies with high-performance scores of 0.96 for the diagnosis of brain tumors using MRI brain scans.

## Discussions

In this research, we have introduced a novel neural network method, CVG-Net, for feature engineering and the timely diagnosis of brain tumors. This proposed method combines 2DCNN and VGG16 to extract spatial features from MRI images. Our experiments utilized a multi-class MRI image dataset comprising 21,672 images related to glioma tumors, meningioma tumors, and pituitary tumors. Our commitment to privacy and confidentiality underscores the implementation of robust data anonymization techniques, thereby mitigating the risk of any inadvertent disclosure of sensitive personal details. This explicit ethical discourse is pivotal not only in upholding the integrity of our research but also in fostering trust within the scientific community and, more importantly, among the individuals whose health data form the foundation of our study.

## Study limitations

In this research study, we have introduced a novel CVG-Net for the prompt diagnosis of brain tumors. Nevertheless, our proposed research does have some limitations. The accuracy score of our approach, currently at 96%, could be further improved by minimizing the loss error rates. To address potential overfitting issues or biases in the machine learning models, we have employed the k-fold cross-validation mechanism, which validates the

performance accuracy scores. Moreover, the deep learning-based neural network models we utilized are computationally expensive and can be optimized by reducing their layered architectures.

# CONCLUSIONS AND FUTURE WORK

This research proposes an effective neural network approach for the timely diagnosis of brain tumors. Our experiments utilized a multi-class MRI image dataset comprising 21,672 images related to glioma tumors, meningioma tumors, and pituitary tumors. We introduced a novel neural network-based feature engineering approach called CVG-Net, which combines 2DCNN and VGG16. The proposed CVG-Net method extracts spatial features from MRI images using 2DCNN and VGG16 without human intervention. The newly created hybrid feature set is then input into machine learning models to diagnose brain tumors. We utilized four advanced machine learning models and two deep learning models based on neural networks for comparative performance evaluations. We also balanced the multi-class MRI image features data using the SMOTE approach. Extensive research experiments demonstrate that utilizing the proposed CVG-Net, the k-nearest neighbors classifier outperformed state-of-the-art studies with a k-fold accuracy performance score of 0.96. We also applied hyperparameter tuning to enhance performance for multi-class brain tumor diagnosis.

## Future work

In the future, we intend to design a graphical user interface app for medical specialists to facilitate the effective diagnosis of brain tumors using MRI scans. Also we plan to enhance the performance by implementing transfer learning-based advanced neural networks. The designed app will utilize the proposed approach in the backend for diagnosis.

## Funding

This research is supported by Princess Nourah bint Abdulrahman University Researchers Supporting Project number (PNURSP2024R136), Princess Nourah bint Abdulrahman University, Riyadh, Saudi Arabia. Prince Sultan University, Riyadh Saudi Arabia supported the Article Processing Charges (APC) of this publication. Princess Nourah bint Abdulrahman University had a role in the study design, conduct, data analysis and interpretation (determined the scope of the work, directed the analysis, gave feedback and dictated the desired output/results verification), manuscript writing, and dissemination of results.

## Grant Disclosures

The following grant information was disclosed by the authors:
Princess Nourah bint Abdulrahman University Researchers: PNURSP2024R136.
Princess Nourah bint Abdulrahman University, Riyadh, Saudi Arabia.
Prince Sultan University, Riyadh Saudi Arabia.

## Competing Interests

The authors declare there are no competing interests.

## Author Contributions

- Shaha Al-Otaibi conceived and designed the experiments, analyzed the data, authored or reviewed drafts of the article, and approved the final draft.
- Amjad Rehman conceived and designed the experiments, prepared figures and/or tables, and approved the final draft.
- Ali Raza performed the experiments, performed the computation work, prepared figures and/or tables, authored or reviewed drafts of the article, and approved the final draft.
- Jaber Alyami analyzed the data, performed the computation work, prepared figures and/or tables, and approved the final draft.
- Tanzila Saba analyzed the data, authored or reviewed drafts of the article, and approved the final draft.

## Data Availability

The dataset is available at Kaggle: https://www.kaggle.com/datasets/mohammadhossein77/brain-tumors-dataset.

The code is available in the Supplementary Files.

## Supplemental Information

Supplemental information for this article can be found online at http://dx.doi.org/10.7717/peerj-cs.2008#supplemental-information.

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
