# Peer review of "CVG-Net: novel transfer learning based deep features for diagnosis of brain tumors using MRI scans"

_PeerJ Computer Science, doi:10.7717/peerj-cs.2008_

## Round 0.1 · original submission · Major Revisions

The reviewers have substantial concerns about this manuscript. The authors should provide point-to-point responses to address all the concerns and provide a revised manuscript with the revised parts being marked in different color.

Reviewer 1 ·

Basic reporting

The manuscript titled "CVG-Net: Novel transfer learning based deep features for diagnosis of brain tumors using MRI scans" by Dr. Alotaibi and colleagues presents a novel approach to the diagnosis of brain tumors using Magnetic Resonance Imaging (MRI) scans. Utilizing a multi-class MRI image dataset, the authors introduce an innovative neural network-based method, combining 2D Convolutional Neural Network (2DCNN) and VGG16, to create the CVG-Net. This network is used to extract spatial features from MRI images without human intervention, and the features are then analyzed using machine learning models. The paper is well-structured, employing a clear, professional language and providing a comprehensive background and literature review. The experimental design is robust, and the findings are presented with a high level of detail, demonstrating a potentially significant contribution to the field of medical imaging and brain tumor diagnosis. Based on the thoroughness and innovative nature of the research, I recommend this paper for publication, albeit with some revisions as noted below.

Experimental design

The use of a multi-class MRI image dataset in conjunction with the innovative CVG-Net approach, integrating 2D Convolutional Neural Network and VGG16, is particularly noteworthy. This novel methodology allows for a detailed and sophisticated analysis of MRI scans for brain tumor diagnosis. However, the design could be enhanced by providing a more detailed justification for the choice of dataset, including its size, diversity, and how it represents the broader population. This would help in assessing the generalizability of the findings. Additionally, a more explicit discussion on ethical considerations, particularly in relation to the use of patient data, would strengthen the study's adherence to ethical research standards.

Validity of the findings

The high accuracy, precision, and recall rates reported provide strong evidence for the efficacy of the proposed method. However, the study could benefit from a more critical examination of potential limitations and biases, particularly in the context of dataset selection and model generalizability. Addressing potential overfitting issues or biases in the machine learning models used would provide a more comprehensive understanding of the method's reliability. Moreover, situating these findings within the broader landscape of existing diagnostic methods, through comparative analysis, would offer valuable insights into the relative strengths and potential improvements of the CVG-Net approach.

Additional comments

Minor comments for improvement involve some aspects of data presentation and clarification. The authors should ensure that all figures and tables are clearly labeled and referenced within the text. A more detailed description of the dataset, including its sources and any preprocessing steps taken, would add transparency and reproducibility to the study. Furthermore, while the language used is generally clear and professional, a few sections could benefit from minor grammatical and syntactical revisions to enhance readability. Addressing these minor points would further polish the paper.

Reviewer 2 ·

Basic reporting

In this study, Alotaibi et al. proposes an neural network-based approach for the diagnosis of brain tumors using Magnetic Resonance Imaging (MRI) scans. The study introduces CVG-Net, a hybrid neural network that combines 2D Convolutional Neural Network (2DCNN) and VGG16 for feature extraction from MRI images, a process carried out without human intervention. The methodology employed includes the use of a multiclass MRI image dataset, feature extraction through transfer learning, and data balancing using the Synthetic Minority Over-sampling Technique (SMOTE). The research demonstrates the effectiveness of the CVG-Net in diagnosing glioma, meningioma, and pituitary tumors, achieving a k-fold accuracy performance score of 0.96. Despite its results, the study acknowledges the need for further performance enhancement, particularly in reducing loss scores during testing, and discusses the implications of these findings for medical imaging and brain tumor diagnosis.

Experimental design

no comment

Validity of the findings

The authors claimed that the CVG-Net has accuracy performance score of 0.96. Since Latif et al. (2022) claim they have 0.96 performance accuracy with SVM, and Kibriya et al. (2022) claim that they have 0.97 performance accuracy with classic neural net work. What is the advantage of using the “Novel” transfer learning process for this task since the computational burden (which is another important point the authors should elaborate more on) will be larger?
3. In section 4.8 regarding table 11, the authors made comparison of state-of-the-art studies for brain tumor diagnosis. The authors should confirm if the same dataset is used for training for the models.

2. For figure 5 and 6, the authors should elaborate more on the figure captions so the reader can get enough information for understanding the manuscript.

3. The loss function and accuracy plots that the author provided in figure 5 and 6 seems not converging, can the authors provide more details on how they decide to stop the training?

Additional comments

The literature summary in Table 1 is very insufficient, a more comprehensive summary is required. A clearer explanation on figure-of-merit should be listed. For example, for Kibriya et al. (2022), why “Classical neural networks were used for classification” is a limitation?

Reviewer 3 ·

Basic reporting

The paper is relatively well-written in professional English, making it not hard to understand.

However, literature review is not good. First, the studies are not grouped into categories and making it hard to follow. Second, at least some of the work the authors mentioned are poorly put into the paper. For instance, Sharif et al. (2020) paper is basically a sentence to sentence rephrasing. Third, comparison of the performance on different metrics and different dataset is not fair.
Authors should completely rewrite literature review and need to remove Table 1.

The methodology is very confusing. From my understanding, the proposed approach is using an emsemble of pretrained VGG16 and a from-scratch CNN, training as four-class classifier on the dataset. Then only use intermediate layers for KNN classifier. However, this is only very vaguely implied, making it hard to understand.
Moreover, this is a very typical use of ensemble techinque. I do not agree it is innovative as authors claimed.

Experiment and results are not relevant enough since there is barely any comparison on the same datasets with other approaches, and is merely an ablation study.

Experimental design

The experimental design, as mentioned above, is not grounded. It is simply an ablation study on each of the component without comparing to other approaches under the same metrics and datasets.

Some of the figures are also not clear, for instance, Figure 5 and 6. And computation time Table 10 is also confusing without comparison with other approaches, it is unclear how exactly these complexity is evaluated either.

Validity of the findings

lack of sufficient and convicing comparison to other methods under same datasets and metrics. Underlying dataset is not provided. Neither methodology nor evaluation support authors claim of revolutionize early brain tumor diagnosis.

---

## Round 0.2 · accepted · Accept

Reviewers are satisfied with the revisions, and I concur to recommend accepting this manuscript.

Reviewer 1 ·

Basic reporting

The authors have addressed all my concerns and I am satisfied with the revised manuscript.

Experimental design

NA

Validity of the findings

NA

Additional comments

NA

Reviewer 2 ·

Basic reporting

The authors addressed my concerns and I have no further comments

Experimental design

NA

Validity of the findings

NA

Additional comments

NA

Reviewer 3 ·

Basic reporting

The updated manuscript has shown significant improvement from the original and most of the issues has been well address. The writing has improved, methods are better illustrated and explained.
I don't have any other revisions.

Experimental design

NA

Validity of the findings

NA